# Outpatient Cancer Care Delivery in the Context of E-Oncology: A French Perspective on “Cancer outside the Hospital Walls”

**DOI:** 10.3390/cancers11020219

**Published:** 2019-02-14

**Authors:** François Bertucci, Anne-Gaëlle Le Corroller-Soriano, Audrey Monneur-Miramon, Jean-François Moulin, Sylvain Fluzin, Dominique Maraninchi, Anthony Gonçalves

**Affiliations:** 1Aix-Marseille Univ, INSERM U1068, CNRS UMR7258, Département d’Oncologie Médicale, Institut Paoli-Calmettes, CRCM, 13009 Marseille, France; monneura@ipc.unicancer.fr (A.M.-M.); Moulinjf@ipc.unicancer.fr (J.-F.M.); maraninchid@ipc.unicancer.fr (D.M.); goncalvesa@ipc.unicancer.fr (A.G.); 2Aix-Marseille Univ, INSERM, IRD, SESSTIM, 13009 Marseille, France; anne-gaelle.le-corroller@inserm.fr; 3Direction du Système d’Information et de l’Organisation, Institut Paoli-Calmettes, 13009 Marseille, France; Fluzins@ipc.unicancer.fr

**Keywords:** outpatient, cancer, e-health, digital, telemedicine

## Abstract

In oncology, the treatment of patients outside of hospitals has become imperative due to an increasing number of patients who are older and live longer, along with issues such as medical desertification, oncologist hyperspecialization, and difficulties in financing mounting health expenditures. Treatments have become less “invasive”, with greater precision and efficiency. Patients can therefore receive most of their care outside of hospitals. The development of e-health can address these new imperatives. In this letter, we describe the different e-health tools and their potential clinical impacts in oncology, as already reported at every level of care, including education, prevention, diagnosis, treatment, and monitoring. A few randomized studies have yet demonstrated the clinical benefit. We also comment on issues and limits of “cancer outside the hospital walls” from the point of view of patients, health care professionals, health facilities, and public authorities. Care providers in hospitals and communities will have to adapt to these changes within well-coordinated networks in order to better meet patient expectations regarding increasing education and personalizing management. Ultimately, controlled studies should aim to definitively demonstrate areas of interest, benefits, and incentives, for not only patients, but also caregivers (formal and informal) and health care providers, health care facilities, and the nation.

## 1. Introduction

During the past few years, anti-cancer treatments have become less “invasive”, more precise, and more effective [1,2,3], while ambulatory care has also improved [4]. At the same time, the digital revolution and its massive repercussions [5,6,7,8,9] are constantly changing our ways of treating cancer and life itself. It has become commonplace to search for information on our smartphones, to make purchases remotely, or to manage our bank accounts from home with complete confidentiality. The number of sites and applications that allow us to record and post information via social networks is also constantly increasing alongside connections between the diverse fields of life.

The unification of these advances in medicine and communication makes it possible to envision a future in which the patient, better informed and connected to their caregivers, will no longer be required to actually stay inside the hospital. However, this perspective creates many questions. Is this future a fairy tale or already a reality? Could we imagine patients receiving the majority of their treatment while at home (remote supervision), or continuing to work “normally” after only a short ambulatory stay for surgery and radiotherapy? Is it desirable? What will the ramifications be for the economy, for the hospital, in particular the health professions in a comprehensive cancer center, and of course, for the patient’s life and relationships with caregivers? In this letter, we depict the current landscape of e-medicine in the field of cancer, and comment on the potential benefits and limits related to the use of e-health in the care management of cancer patients.

## 2. Changes in Cancer Care

### 2.1. Epidemiological and Economic Constraints

Cancer is a major public health issue in developed countries. In France, despite improvements in the standardized incidence and mortality rates [10], the burden of cancer remains heavy and its incidence is increasing due to demographic growth and aging. Furthermore, patients are now living longer thanks in part to treatment advances, with metastatic diseases now becoming “chronic” due to more tolerable treatments, along with an increase in the number of “cured” patients requiring post-treatment follow-up. Meanwhile, the patients themselves are better informed and increasingly eager for information, thereby becoming more and more involved as actors in both the decision-making process and treatment. Their demands and those of their relatives have increased regarding the quality of diagnostic notifications and communication, access to innovation, attention to quality of life, psychological support, and socio-professional rehabilitation.

At the same time, general practitioners and pharmacists feel insufficiently trained for and uncomfortable with this constant therapeutic evolution. All of these changes have led to repercussions ranging from a saturation of specialized health care establishments, to oncologists seeing unending growth in the number of active patient files. The rapid evolution of knowledge and increasing treatment complexity subsequently impose both continuous training and hyperspecialization of medical practitioners, resulting in personnel shortages in certain geographic regions and medical fields. Cancer centers find consultation rooms, beds, operating rooms, and radiotherapy machines increasingly saturated. Therefore, the effective care of patients outside of hospitals has become a public health imperative.

It is also an economic imperative due to difficulties in financing the ever-increasing costs of health care. Of the 7.25 billion euros spent in 2012 in France, hospital stays and sessions in health care establishments (including transportation) represented 64.5% of expenditures. Hospital stays also increase economic cost in terms of professional absenteeism and decreased productivity. As such, the development of ambulatory or at-home treatments is desperately needed.

### 2.2. Medical-Scientific Advances and Changes in Cancer Management

The French EVOLPEC study “Quelle prise en charge des cancers en 2020?”, conducted by UNICANCER (http://www.unicancer.fr) on cancer management in 2020 in French cancer centers, has identified five main structural changes that should help promote the management of cancer patients “outside the walls” [11]. The first change concerns ambulatory surgery, of which development has been thoroughly encouraged by public authorities because of the expected benefits for patients, health facilities, and insurance organizations. In 2014 in France, ~26% of cancer surgery was performed on an outpatient basis, versus ~17% in 2010. However, ambulatory surgery requires a complex logistical organization for postoperative monitoring at home. The second structural change is hypofractionated radiotherapy, made possible by the increasingly precise delivery of radiation and corresponding increase in dose per fraction. It represents a solution to both improving access to care and increasing the effectiveness/toxicity ratio, but requires a longer preparation time. In the U.S., ~35% of women over 50 years of age with breast cancer were treated with hypofractionated radiotherapy in 2013, compared to ~11% in 2008 [12]. Thirdly, there should be an important change in the development of oral and targeted chemotherapies. The latter are most often administered outside the hospital, orally, continuously, and indefinitely or “for life”, which raises issues regarding compliance and monitoring, as there are more diverse and complex toxicities than those found in traditional chemotherapy. Although this requires less of a physical presence in hospitals, it requires lengthier consultations for the education of patients concerning the toxicities and risks of drug interactions, as well as new programs of city-hospital therapeutic education [13] and of regular, prolonged, and specialized remote surveillance. The fourth structural change concerns the continued development of the molecular characterization of tumors, such as in the collection of patient blood at home for next-generation sequencing and analysis of circulating tumor DNA. The fifth and final change is the development of interventional radiology, allowing for extremely precise, yet less invasive diagnostic and therapeutic, procedures. According to EVOLPEC study, the number of hospital stays for interventional radiology should increase by a factor of four by 2020, which would allow for a reduction of surgical stays by 5%, with one-third of the procedures carried out on an outpatient basis.

Thus, patient management in cancer centers is shifting towards a decrease in the number and duration of hospital stays, with a succession of specialized and increasingly complex multidisciplinary interventions carried out on an outpatient basis at predefined times. The amount of at-home care and follow-up will increase as well, which will in turn decrease face-to-face contact with an oncologist. The role of the hospital will be less centered on hospital stays and will instead focus on the direction and coordination of disciplines and technologies necessary for care and treatment. This new structural organization will allow for remote, “outside the walls”, support that ensures continuous and prolonged care and follow-up in a safe and equitable manner. It requires the development of therapeutic education for patients and care providers, city-hospital cooperation, and new tools for information and communication.

## 3. E-Health

This ambulatory shift coincides with and will greatly benefit from the ongoing digital revolution and the corresponding novel technologies developed in information and communication via the Internet. Currently, internet use revolves around mobility (smartphones, and tablets), connecting to every-day objects (watches, vehicles, houses, etc.), and data management. Internet access is now available almost anywhere, at any time, and through any number of devices. This number in France is expected to reach nearly two billion devices by 2020. They have already made inroads into the realm of health care, with 13% of French people currently equipped with an e-health device. E-health is defined as “the application of information and communication technologies to all activities related to health and the provision of health care remotely” [14,15]. In developed countries, e-health is considered to be “an opportunity to improve the efficiency of the health care system” [16]. E-health includes many different components that can be used for many different applications in oncology (Figure 1). For instance, telemedicine can improve access to information, screening and diagnosis, treatment and follow-up by remotely transmitting medical data (patient, radiological, pathological images, or videos) to a medical appointment, consultation, or even a surgical procedure. It abolishes distances between patients and hospitals, and between advanced, resource-rich and remote, resource-poor care centers [17,18], ultimately improving both the quality and equity of health access in distant and underserved geographic areas, as well as contributing to the training of care providers [19] and the removal of transport costs. E-learning includes remotely delivered training services for health care providers and patients about diseases and treatments, and can use “serious games” as a more ludic and interactive option. Meanwhile, “quantified self” tools allow patients to collect their own medical data in order to improve health and well-being. These include connected and smart objects (wristbands, watches, glasses, clothes, etc.) that measure health parameters (pulse, blood pressure, glycaemia, etc.), which are transmitted via mobile applications and ultimately sent to caregivers for follow-up and management optimization.

M-health is defined as the use of mobile communications (voice or short message service, SMS) and devices such as smartphones and tablets (mobile applications, localization systems, mobile internet, etc.) in the health domain [20]; in 2015, ~40,000 health-dedicated apps were already available in Apple or Google stores and the market was expected to reach 10 billion dollars in 2018. Due to their permanent connectivity, these tools may allow for improved personalized support, follow-up, and therapeutic management. Phone sensors are also able to repeatedly collect data at any time, being a much more reliable data source than any tool previously used. Sensors can provide quantitative information that is usually operator-independent and (depending on the placement and on the proper use) subject-independent. For example, some apps are able to send automatic messages to remind patients to take oral treatments [21,22,23]. Apple has even developed two open source frameworks for creating iPhone applications that are dedicated either to research (ResearchKit) or to daily patient monitoring (CareKit) (http://www.apple.com/fr/researchkit/). ResearchKit apps facilitate patient recruitment and the corresponding data collection in clinical trials. Another component of e-health is “Big Data”, in which large volumes of digital medical data from public or private databases are used and analyzed in order to improve health systems and research [24]. It is estimated that 30% of currently stored data are health-related [16], a large part of which could hold helpful insights for patients, health care systems, and research.

## 4. Potential Impacts of E-Health in Oncology

E-health has many potential clinical impacts in oncology, and has already begun to affect every level of care, including education, prevention, diagnosis, treatment, and monitoring. The primary studies, mainly randomized clinical trials, are summarized in Table 1.

### 4.1. Access to Information and Education

From the patient perspective, the lack of general information on diseases and treatments is a major issue concerning participation in the decision-making for and adherence to treatments [35,36]. Websites, social networks, forums, and smartphone applications could solve this issue as they enable immediate access to unlimited information [37,38]. A recent study on Apple’s iTunes store found 28 applications dedicated to general information on cancer, out of 77 “cancer” applications identified [39]. These tools also help patients better manage their diseases by providing flowcharts of scheduled exams, appointments, and biomedical exam results. They are also helpful and effective in improving the quality of life of patients; for example; a randomized controlled trial showed that web applications- and text message-based patient education in Mohs micrographic surgery reduced the preoperative anxiety of patients [25].

Access to information is also important for oncologists in routine practice, and these allow for rapid access to the most recent and up-to-date data, guidelines, drug lists, and corresponding toxicities. [40,41]. For example, a pilot study showed the potential of smartphone application for improving the knowledge of colorectal cancer screening among 50 internal medicine residents [42]. Web-based educational materials can help address the needs of oncology health care professionals seeking to understand up-to-date treatment strategies. A controlled trial enrolled 751 participants who had previously taken a learning style survey: participants enrolled in the intervention group viewed educational materials consistent with their preferences for learning (reading, listening, and/or watching), and participants in the control group viewed educational materials typical of the My Cancer Genome website. Educational materials covered the topic of treatment of metastatic breast cancer using CDK4/6 inhibitors. The intervention arm showed greater improvement in post-test score and a higher follow-up test score than the control group [26], suggesting more learning with web-based learning style-tailored educational material.

### 4.2. Prevention

Websites inform patients about potential risk factors and also help with modifying exposure. Smartphone applications can repeatedly deliver proactive and discreet information and advice anywhere and at any time, immediately attracting the user’s attention and prompting for urgent responses as necessary according to the context.

Thus, in the prevention of cutaneous cancers, several applications have been developed and will be [43,44] or have been tested in clinical trials [27,45]. For example, a randomized clinical trial evaluated a mobile application’s real-time sun exposure protection capabilities via the delivery of advice (protective procedures, risk of sunburn, etc.) and alerts (applying or re-applying sunscreen, ending exposure after a defined time-limit, etc.), in accordance with the current ultraviolet (UV) index, local time, and geographic location [27]; individuals in the experimental arm had improved protection to exposure. A future trial [43] will enroll approximately 60 construction workers across the United Kingdom. This randomized control crossover trial will test the intervention based on text messaging in combination with a supportive smartphone application. The intervention aims to both reduce UV exposure during months with higher UV levels and promote appropriate dietary changes to boost vitamin D levels during months with low UV levels. Such study will provide important information about the effectiveness of a technology-based intervention to promote sun safety and healthy behaviors in outdoor construction workers.

Web-based, informed decision-making tools have created new avenues for helping smokers desiring to quit [46]. In a randomized clinical trial enrolling current smokers desiring to quit, the 6-month rate of biochemically verified tobacco cessation was two-fold higher in the experimental group that received periodic, motivational text messages on their smartphones when compared to the control patients who received irrelevant messages [28]. Further applications dedicated to breast cancer prevention [47,48], and assistance with alcohol withdrawal [49] are currently under development as well.

### 4.3. Screening and Diagnosis

Screening may also be improved by e-health [50], notably for cervical cancer [51]. In Tanzania, a screening program dedicated to cervical uterine carcinoma was set up using smartphones. Nurses located in the most distant regions of the country used their smartphones to take pictures of cervices and send them by multimedia messaging service (MMS)to physicians in a cancer center. The physicians would then respond via text message with the appropriate actions to take. Connected2Care is a multicenter, randomized trial launched in Tanzania [52]: 700 women testing positive to high-risk Human Papilloma Virus (HPV) will be randomly assigned to the mobile phone-based Short Message Service (SMS) intervention group or the control group (standard care). In a period of 10 months, the intervention group will receive 15 one-directional health educative text messages and SMS reminders for their appointment. Primary outcome is attendance rate for follow-up. Other examples of web-based screening of cervical cancer in low- and medium-income countries come from Madagascar [53] and rural Indian zones [54]. Such a procedure could help prevent these frequent cancers by eliminating the lengthy travel necessary for examinations, and it was for this reason that a similar program of telecolposcopy was set up at eight spoke sites across Arkansas [55]. Mobile applications have also been developed for improving the colon cancer screening [56].

Diagnosis may also benefit from e-health tools that allow for remote clinical examination, as recently reported for non-invasive detection of anemia using only patient-sourced photos [29]. In oncology, certain mobile applications have been developed to help diagnose cutaneous cancers [57,58] that use smartphones to take photos of cutaneous lesions, which then undergo software-assisted image analysis [59] or are transferred via the Internet to a dedicated dermatologist. In the US, Google has launched a novel “Symptom Search” feature aimed at refining symptoms analysis that also sends disease information to patients and suggests certain medications automatically, with a statement that it is not a substitute for consulting a health care professional. Regarding the pathological diagnosis, a recent study showed that easy, fast, and high-quality image capturing and transfer are possible from cytology slides using smartphones, with high intraobserver Kappa agreement (84.3%) between the microscopic cytopathological diagnoses and remote smartphone image diagnoses [30]. Smartphone-based molecular analyses are also being developed [60,61,62].

### 4.4. Treatment

E-health could be a major asset in facilitating treatment “outside the walls”, by aiding in improved compliance, toxicity management, and earlier discharge from hospitals.

Low compliance with oral treatments, such as hormonal therapy in breast cancer or imatinib in gastro-intestinal stromal tumors (GIST), results in loss of efficacy [63,64]. To improve compliancy, many applications have been developed [65], which automatically send daily motivational messages, reminding patients to take their medications at the right times [66,67]. Clinical trials are ongoing regarding hormonal therapy of breast cancer patients for example [68,69].

Meanwhile, cancer treatment toxicities negatively affect quality of life and may have fatal outcomes when detected too late, and yet patients are still rather reluctant to report side effects to their physicians [70], and a posteriori reporting underestimates the actual severity [71]. Patient education, fact sheets, patient notebooks, and improved communication are known to be effective tools in promoting early reporting of toxicities, and can be improved even further by e-health technologies. For example, a recent randomized study suggested the feasibility and potentiality of the use of smartphone mobile games for patients with breast cancer receiving chemotherapy. Education using a mobile game led to better patient education, improved drug compliance, decreased side effects, and better quality of life compared with conventional education [31]. In addition, patients can self-evaluate their own daily side effects and vital signs and transfer the results via smartphone to health care providers [72]. Then, according to a predefined severity of reported symptoms, an alert is sent to a nurse who contacts the patient and possibly the referring physician. The feasibility of this Patient-Reported Outcomes (PROs) approach was well-demonstrated, with the majority of patients feeling reinsured and considering this procedure to be simple, easy to use, and helpful in quickly resolving their problems [73,74]. PROs are commonly included in cancer clinical trials and the U.S. NCI has developed a PROs version of the Common Terminology Criteria for Adverse Events (PRO-CTCAE) that is currently being evaluated by stakeholders, including the Food and Drug Administration (FDA) [75]. A Norwegian randomized study enrolling breast cancer patients currently under treatment [76] revealed that, when compared to standard management, internet-based support resulted in superior outcomes in terms of both physical and psychological symptoms. In an American prospective trial, 358 patients beginning chemotherapy were randomized to the Symptom Care at Home (SCH) intervention or enhanced Usual Care (UC). Participants called the automated monitoring system daily to report the severity of eleven chemotherapy-related symptoms. SCH participants received automated self-management coaching and nurse practitioner telephone follow-ups for poorly controlled symptoms. The SCH dramatically improved symptoms [32]. Another randomized trial compared monitoring symptoms weekly using PRO tablet computers versus traditional care at a physician‘s discretion in 766 patients receiving outpatient chemotherapy for advanced solid tumors. Twelve common symptoms were specifically monitored and e-mail alerts were sent to nurses in case of worsening or severe clinical signs. Changes in health-related quality of life at six months, the primary endpoint, significantly favored the intervention group, with greater improvement (34% versus 18%) and less worsening (38% versus 53%). There were also significantly fewer admissions to hospitals or emergency rooms (−7%) in the intervention group, with a trend towards better overall and quality-adjusted survivals [33].

For the follow-up of patients under targeted therapies, several studies have shown that general practitioners are generally uncomfortable with the situation and that the patients themselves had greater trust in their oncologist. In addition, follow-up consultations are relatively cumbersome and costly for the patient (transport, wait time, stopping work, etc.); yet, if these consultations were well-organized, they could be performed via the Internet, making it possible to drastically reduce the number and cost of face-to-face specialized consultations for cancer patients with a stabilized disease under chronic treatment.

Web-based applications may also improve the postoperative outcome and facilitate earlier discharge from hospital. A randomized controlled trial evaluated a postoperative web-based application intervention to provide real-time symptom monitoring among patients with suspected gynecological cancer who had open bilateral salpingo-oophorectomy surgery [77]. The study established feasibility, acceptability, and some potential benefits of such an approach for gynecological oncology postoperative care.

### 4.5. Post-Treatment Follow-Up

Follow-up consultations aim for the early detection of relapses and management of possible persistent or late drug toxicities, and also identify disease-related psychological, social, and professional problems in patients’ and families’ lives. Unfortunately, general practitioners feel poorly trained in these areas while patients prefer trusting their oncologist. However, for oncologists, the average duration for consultations has not changed, even though the number of consultations is constantly increasing [78]. Access to these consultations in cancer centers may also be problematic for patients that live far from the hospital or have transportation difficulties, which likely contributes to the relatively worse outcomes of patients living in rural areas or underserved suburbs [79,80]. E-health could solve these issues as well via virtual consultations with oncologists. Currently, Skype is the most frequently used method for this, and has been shown by numerous studies to be satisfactory for both patients and physicians [81], with potential clinical benefits. For this reason, a large prospective study is being conducted in Great Britain to evaluate virtual consultations [82]. In the cancer domain, a review [83] of the role of technology in patient follow-up evaluated thirteen randomized studies, most of which used “low-tech” approaches, such as phone calls by nurses, with only two studies based on either managing treatment toxicities with a mobile application [84] or evaluating quality of life with a personal electronic agenda [85].

Digital follow-up may also improve survival. Indeed, in spite of routine surveillance based on regular clinico-radiological exams every 4, 6, or 12 months, disease recurrences frequently occur outside of scheduled visits. A recent study illustrated that a mobile application-driven follow-up could improve patient outcomes in lung cancer [34]. The authors designed an e-follow-up application (e-FAP) to provide individualized imaging schedules based on patient self-evaluations of clinical symptoms. Two prospective studies have already demonstrated the reliability of e-FAPs, with relapses detected (on average) five weeks earlier than with routine scheduled imaging [86,87]. A pilot study has suggested an improved one-year survival rate in the e-FAP arm compared to that in the retrospective control arm [88]. A prospective multicenter randomized trial [34] tested the hypothesis that a web-mediated follow-up (experimental arm) would improve the overall survival (OS) in lung cancer patients with a high risk of relapse or progression compared to that in patients with a routine follow-up (control arm with computerized tomography (CT)-scans scheduled every three to six months according to the disease stage). In addition, in the experimental group, patients were offered use of the mobile application “Moovcare” to record via computer, tablet, or smartphone the status of twelve symptoms (fatigue, cough, dyspnea, pain, loss of appetite, fever, etc.), which could indicate a possible relapse. After data transmission, an algorithm generated a relapse score based on the association and evolution of symptoms, which would email an alert to the oncologist, who would then move up appointments for CT scans and consultations. A total of 121 stage III/IV lung cancer patients were randomized and analyzed. The trial was stopped prematurely after an interim analysis revealed that the experimental arm had a more improved median OS (19 months) than the control arm (12 months). The respective one-year OS rates were 75% versus 49%. The relapse rate was similar in both arms (51% versus 49%), but performance status at initial relapse was higher in the experimental arm, which allowed more patients to receive optimal treatment. Quality of life was also better and the average number of CT scans per year and per patient was lower in this group.

E-health technologies may also improve the quality of survival. They promote emotional well-being in breast cancer patients within the three months of diagnosis [89]. Whether they may also reduce the fear of recurrence among breast cancer survivors is being tested in a Japan randomized controlled trial, the SMILE project (SMartphone Intervention to LEssen fear of cancer recurrence) [90]. The feasibility, validity and reliability of smartphone to measure physical activity and fitness in patients with cancer have been demonstrated [91,92]. Such an approach is well accepted by patients [93] and results in increase in physical activity and capacity and treatment-related symptoms even during active chemotherapy [94,95]. A mobile application (OncoFood) that assesses and evaluates dietary behaviors in oncologic patients was tested in a pilot study [96]: the application group gained significantly more weight than the control group, and the skeletal muscle mass showed a significant increase.

## 5. Issues and Limits of “Cancer Outside the Hospital Walls”

Cancer care “outside the walls” is becoming a critical issue and will have human, economic, and organizational consequences (Table 2). Besides the expected benefits, several questions and fears are emerging [15].

### 5.1. For the Patients

Two crucial points for patients are autonomy and quality of life. Patient satisfaction is high with ambulatory surgery [11] and chemotherapy at day hospitals or at home [97,98]. Remote consultations and treatments at home or at work offer numerous advantages: greater comfort, less time wasted in transportation and stressful waiting rooms, improved patient involvement in treatment, relative de-dramatization of disease, greater equity in relationships with caregivers, and easier access to care for disabled patients or for those living in distant geographical areas. Digital tools also provide easier and more rapid access to health care professionals, the latter being able to react quickly for improved orientation, information, education, and support for patients. Several studies, although small in size, revealed high patient satisfaction rates for e-oncology approaches [99]. Digital tools facilitate access to information, making it instant, unlimited, and possible to be shared with other patients through forums or even “second opinion” websites. 

However, evaluation of mobile applications is crucial [100,101], since negative effects could arise from certain applications. For example, regarding the diagnostic applications for cutaneous cancers, several studies have pointed out significant drawbacks, such as a lack of updates and medical scientific validation [102,103]. The involvement of academic societies as well as regulatory agencies is crucial for guaranteeing patient safety [104]. Regarding the sending and receiving of clinical images with smartphone, the current practices are insufficient to comply with professional and legal obligations, and increase practitioners’ vulnerability to civil and disciplinary proceedings. Further education, realistic policies and adequate software resources are critical to ensure protection of patients and practitioners [105].

There are also questions regarding the risk of patient morale and isolation. Another important challenge is the need for physical examination, as it is more difficult to build an atmosphere of trust during remote consultations and the examinations are of inferior quality. However, when physical examinations are indispensable, they can be easily performed by another physician on-site.

Furthermore, the main components of a successful patient–physician relationship, such as dialogue and active listening, should remain unaffected by digital tools, while caregiver availability, another important component, could actually be increased. Using a tool such as Skype at home could also help improve patient comfort and self-confidence when communicating with their physician, as the environment would be less hostile than a consulting room at a hospital. In addition, a Skype conversation at home could actually increase trust between patients and physicians by improving communication and frequency, as demonstrated in a palliative setting [106].

A further potential limitation of e-health is the digital divide, i.e., certain categories of patients (elderly, fragile, foreign, poorly educated, rural, etc.) have difficulties in accessing the Internet or do not have the necessary broadband connections (i.e., for multimedia communication or large file transfers). In the U.S., the main causes of the digital divide, which may amplify existing inequalities in access to health care, have been analyzed [107] and corrective actions undertaken [15,108].

### 5.2. For Health Care Professionals and Informal Caregivers

Managing cancer patients “outside the walls” with the use of digital tools provides improvements in: medical and paramedical time management, assistance with personnel training, patient education, and medical decisions, access to professional resources, and information exchanges between health care providers inside and outside of the hospital. Altogether, these transformations may create new health professions, especially regarding the coordination of nurses in charge of patient follow-up and of hospital and community care. For hospital oncologists, current standard practice will certainly change; some functions will cease to exist as the use of digital tools becomes more frequent. Recent studies reported a high level of oncologist satisfaction with teleoncology consultations [99] or remote supervision of chemotherapy by a referred care center [109].

Even if the benefit of home care for patients in terms of well-being and quality of life is admitted, to date the use of telemedicine tools aimed at the formal (health care professionals) and informal (family members, close friends, etc.) caregivers of cancer patients remains poorly defined [110]. However, the role of informal (and formal) caregivers is of key importance in the ultimate results. Their efforts to care their loved ones have considerable physical and psychological impacts on them, notably with cancer patients [111]. Cancer caregivers need information to manage patients’ symptoms and improve their knowledge in medical procedures to counter their fear of inadequacy. In this context, e-health tools are able to respond to these unmet needs of formation and direct interaction with healthcare professionals [112]. A systematic review [110] regarding the use of telemedicine tools for informal caregivers implemented in cancer care reported significant improvements in some of measured outcomes, but concluded that we are in an exploratory phase and that more detailed and targeted research hypotheses are still needed.

### 5.3. For Health Facilities

Efficient human and material resource management is another major issue that the shift to digital could improve, such as with the optimization of medical schedules and equipment usage, assistance in prescribing medications, managing appointments, and the sharing of medical records A novel and complex organization should emerge on the basis of new digital tools, with greater dependency on effective coordination between communities and hospitals rather than traditional hospital stays, aiming to systematically guarantee a high-quality continuation of patient management from hospital to home. With e-health in cancer centers, the traditional and heavy emphasis on “care” could decrease in favor of prevention, support, and education. Decentralization of some care functions could also allow hospitals to focus on extremely specialized and non-transferable health technologies, while the massive “Big Data” transfer of clinical and biological knowledge should favor the development of precision medicine with more personalized treatments, ultimately leading to a more judicious usage of available drugs, which in turn should reduce costs. Such transfer could be viable also toward very small institutions and agencies (and even small associations of general practitioners), thanks to the increasing use of cloud-based solutions.

### 5.4. For the Public Authorities

There are huge economic and social stakes associated with the development of ambulatory management. However, many questions remain regarding the impact of an “outside the walls” approach to cancer management, especially in terms of the digital revolution. Will it lead to reduced health expenditures or only transfer them from the hospital budget to other community actors (general practitioners, community nurses, patient families, etc.)? Will it be an economically viable solution for hospitals? Will it be possible to allocate the gain in resources to novel professions or organizations? How will pricing be determined, and what will be the subsequent methods of reimbursement for hospitals? This latter parameter is a key issue, since it may provide either strong incentives or disincentives for the orientation of hospital strategies. This issue may be best illustrated by the evolution of the pricing mechanisms for outpatient surgery in France. Since 2003, the procedure-based invoicing system has become an incentive for ambulatory surgery, with an expected increase in hospital revenues due to the potential for treating larger volumes of patients when compared to inpatient surgery.

In addition to the possibility of increasing revenue via a higher volume of patients, a further incentive could be achieved by modifying other pricing methods. In the case of hypofractionated radiotherapy, a payment-per-session method penalizes this innovative and more patient-convenient approach as there are fewer sessions, resulting in fewer reimbursements and decreased revenue for hospitals, even though each session is longer and the whole procedure is more time-consuming due to longer preparation times. In this case, a possible solution could be flat-rate reimbursements. A similar question can be raised for outpatient oral chemotherapy, as a recent study [113] indicated that a significant decrease in potential hospital revenue could occur because of transference to other actors. This potential transfer and the resulting financial losses could incite hospitals to favor intravenous chemotherapy in day hospitals in order to maintain their reimbursements.

Thus, the cost of outpatient management in cancer treatment must be analyzed according to the specific perspective of the different actors: hospitals, health insurance systems, and the national community. From the hospital perspective, the key point is the distribution and accumulation of resources needed for optimal patient management; for example, according to the French National Study of Costs (Etude Nationale des Coûts, ENC), subtotal mastectomies performed in 2012 for malignant tumors in the public sector [114] represented an average cost of 2384 € (outpatient) or 3467 € (inpatient). Therefore, outpatient surgical breast cancer management is clearly associated with a reduction for hospital in accumulated resources via reimbursement. Similarly, a French study has shown that outpatient administration of chemotherapy, with monitoring and follow-up at home by phone, had a cost (without toxicity) of 3858 € versus 8431 €, respectively, for inpatient administration, while in cases where toxicity required hospitalization, costs were 7989 € versus 17,572 €, respectively [115]. Therefore, from the perspective of health insurance systems, an ambulatory shift should be highly beneficial. Indeed, the reimbursement tariffs in France are based on hospital operating costs and are thus periodically reevaluated. A decrease in hospital operating costs will ultimately result in a decrease in reimbursement tariffs. However, a complete analysis of the resources distributed to the overall community (general practitioners, nurses, etc.) from outpatient management could mitigate the financial advantage for health insurance systems. At the national level, it remains to be seen whether outpatient management will ultimately translate to financial benefits. The main issue here is the possible transfer of costs associated with an ambulatory shift, as outpatient management may sometimes require the presence of a companion at home, whose work and time are associated with a specific cost that is not currently taken into account by most calculations. Therefore, the ambulatory shift might essentially transfer costs previously assumed by the hospital to other actors (families, general practitioners, etc.). Only economic studies, conducted from the national perspective that include each and every resource needed for outpatient management, will allow for a definitive conclusion on its putative advantage compared to that related to conventional hospitalization. Of note, any financial advantage identified should also be compared to the procedure’s efficacy, including patients’ quality of life.

A recent systematic review showed that the costs of home-based telemedicine programs varied substantially by program components, disease type, equipment used, and services provided [116]. The selected studies indicated that home telemedicine programs reduced care costs, although detailed cost data were either incomplete or not presented in detail. A comprehensive analysis of the cost of home-based telemedicine programs and their determinants is still required before the cost efficiency of these programs can be better understood, which becomes crucial for these programs to be more widely adopted and reimbursed. Regarding teleoncology, cost-effectiveness studies are limited. Two studies have been conducted evaluating the costs of telemedicine in cancer patients from rural areas living far from their designated care centers, which revealed economic cost reductions, notably in terms of transportation and housing [117,118].

## 6. Conclusions

Accelerated by new digital communication tools, a revolution with the potential to erase inequalities in care quality and access is ongoing, prompting cancer patients to leave hospital walls behind. It will have tremendous human, economic, and organizational impacts, and will require profound modifications in the roles of health care professionals with priority given to patient information, education, and support. Care providers in hospitals and communities will have to adapt to these changes by working within well-coordinated networks in order to better meet patient expectations regarding increasing education and personalizing management. Ultimately, controlled studies should aim to definitively demonstrate areas of interest, benefits, and incentives, not only for patients, but also for caregivers (formal and informal), health care providers, health care facilities, and the nation.

## Figures and Tables

**Figure 1 cancers-11-00219-f001:**
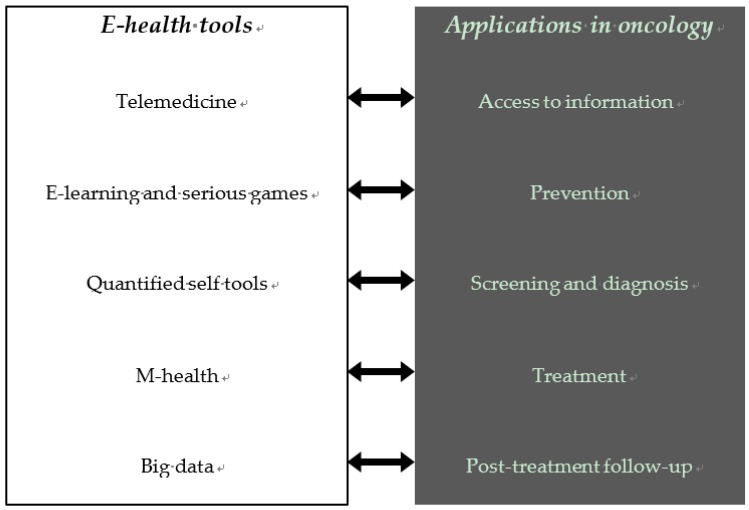
E-health and applications in oncology. The left panel contains different e-health tools and the right panel includes different applications of e-health in oncology.

**Table 1 cancers-11-00219-t001:** Examples of studies published.

Field	Study Type	Cancer or Subjects Type	Number of Subjects	E-Health Tools Involved	Main Results of the Experimental vs. Control Arms	Reference
Information Access	Randomized clinical trial, 4 arms (web applications vs. text messages vs. both vs. standard care)	Cutaneous cancer before Mohs micrographic surgery	90	Web applications and text messages for patient education	Reduction of patients’ preoperative anxiety	[25]
Educating Health care professionals	Controlled clinical trial with concealed allocation, 2 arms (educational materials consistent with subjects’ preferences for learning vs. materials typical of the My Cancer Genome website)	Oncology health care professionals	751	Web-based	Improved learning with tailored, web-based learning style educational material	[26]
Prevention	Randomized clinical trial, 2 arms (mobile application vs. control)	Adults from the Knowledge Panel, 18 years or older who owned an Android smartphone	604	Mobile application providing personalized, real-time sun protection advice	Improved sun protection	[27]
Prevention	Randomized clinical trial, 2 arms (mobile application vs. control)	UK smokers willing to attempt quitting	5800	Periodic, motivational text messages on smartphones	Higher 6-month rate of biochemically-verified tobacco cessation	[28]
Diagnosis	Prospective development of a non-invasive anemia screening tool	Patients with anemia of different etiologies and healthy subjects	337	Smartphone application and photos	Detection of anemia with an accuracy of ±2.4 g/dL (0.92 after personalized calibration) and a sensitivity of 97% when compared with blood count hemoglobin levels	[29]
Diagnosis	Retrospective assessment of smartphone usage in telecytology	Different cytological materials	172	Smartphone photos transferred via WhatsApp^®^	High intraobserver Kappa agreement between microscopic diagnoses and smartphone image diagnoses; change in patient management in 11.4% of cases	[30]
Treatment observance and tolerance	Randomized clinical trial, 2 arms (mobile games vs. standard care)	Patients with metastatic breast cancer planning to receive chemotherapy	76	Smartphone-based mobile games	Better patient education, improved drug compliance, decreased side effects, and better quality of life	[31]
Treatment tolerance	Randomized clinical trial, 2 arms (automated home monitoring and follow-up vs. standard care)	Patients beginning chemotherapy	358	Symptom Care at Home (SCH) intervention	Reduction of clinical symptoms	[32]
Treatment tolerance	Randomized clinical trial, 2 arms (Patient-Reported Outcomes (PROs)-based symptom monitoring vs. standard care)	Patients receiving outpatient chemotherapy for advanced solid tumors	766	PRO tablet computers	Improvements in health-related quality of life at 6 months, fewer admissions to hospitals or emergency rooms, better overall and quality-adjusted survivals	[33]
Follow-up and survival	Randomized clinical trial, 2 arms (e-FAP-based follow-up vs. standard follow-up)	Patients with stage III/IV lung cancer	121	E-follow-up application (e-FAP)	Improved overall survival (median and 1-year overall survival); similar relapse rates, but better performance status at initial relapse, and better quality of life	[34]

**Table 2 cancers-11-00219-t002:** Strenghts, Weaknesses, Opportunities, and Threats (SWOT) scheme of application of e-health in oncology.

**Strengths**	Patients - More “actors in healthcare” for oral treatments - Downplay dramatization of diseases and treatments - Improved comfort/quality of life at home and work - Decreased time spent in transportation and “scary” waiting rooms - Greater autonomy in managing appointments - Greater equality in caregiver relationships - Broader and more rapid access to: medical files, second opinions, and disease and treatment information - Sharing of disease and treatment-related experiences (social networks, and forums) - Equal access to care Oncologists and Hospitals - Decision-making support (diagnosis, and treatment) - Information exchange between a city and rural medical centers - Optimization of medical resources: improved time management
**Weaknesses**	- Novel and complex organization - Lack of coordination between healthcare professionals - Insufficient training of non-hospital personnel (doctors, pharmacists, nurses, etc.) with no current method of reimbursement - Insufficient digital training - “Bringing cancer back to home or work”, which could place further strain on familial and professional relationships (loss of confidence and trust) - Care for unsupported companions
**Opportunities**	- Health care cost reduction - Increased cooperation between health care facilities (hospitals, cancer centers, rural health care providers, etc.) - Creation of new health care professions (coordination, and follow-up: nurse navigators) - Digital market
**Threats**	- Patient–caregiver estrangement: feelings of loneliness and anxiety concerning the diseases and treatment toxicity - Virtual “less human” relationships - Trivialization of the burden of care - Poor grasp of the risks involved - Failure to comply with oral treatments - Digital divide (elderly, poorly educated, and foreigners) - Overbooking doctors (burnout) - Elimination of certain hospital functions

Lines are to separate items related to patients (-more actors… etc.… until equal access to care) and items related to oncologists and hospitals.

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
