# Peer review of "Outpatient Cancer Care Delivery in the Context of E-Oncology: A French Perspective on “Cancer outside the Hospital Walls”"

_cancers, 2019, doi:10.3390/cancers11020219_

Reviewer 1 Report

The work described in the paper appear to be well balanced and definitively may represent an interesting review for the oncology area, although what reported could also be of values in other long-term patologies’ treatment. The “Outside the hospital walls” is of course *the* perspective in healthcare (at least in developed countries), for the several reasons thoroughly described in the work.

For all the reasons above, I do believe that the paper should be published with some (really) minor corrections.

The most important one is the following: I suggest the authors to refresh the scientific references (there are only a couple of 2017 and no newer), because in the last couple of years there have been published other interesting studies that might increase the thoroughness and soundness of the present review, without heavily affecting it.

I will also suggest to take a bit more into consideration the role of caregivers with respect to the home treatments, in particular when discussing about e-health services, for their role could be of key importance in the ultimate results.

Coming to the actual paragraphs, I would suggest the authors to clarify a bit further the sentence @ lines 138-141, for the reliability of sensors as data sources cannot be based on their “permanent connectivity”. Rather, sensors could provide quantitative information that is usually operator-independent and (depending on the placement and on the proper use) subject-independent.

Another sentence to further clarify regards the lines 336-337, when introducing the “massive ‘Big Data’ transfer of clinical and biological knowledge”. The overall meaning of the sentence is almost clear, but it may be important to underline that such transfer could be viable also toward very small institutions and agencies (and even small associations of GPs) thanks to the increasing use of cloud-based solutions

There are also a couple of small typo (line 288 erratic “.” , line 316 “provides for”-> “provides”), and a final small suggestion of word substitution (line 354 “had” -> “achieved” or “obtained”).

Reviewer 2 Report

The authors of the article “Outpatient cancer care delivery in the context of e-Oncology: a perspective on “Cancer outside the hospital walls” report on how e-health can address challenges of cancer treatment in an aging society.

I have objectives regarding the method, as the authors rather present a letter or comment than a review. There is no clear rationale and theoretical framework behind the study. Also, the conclusion of the study does not present any new or exciting revelation. So, it is not clear what the article adds to the existing literature on the topic of e-Oncology.

The authors focus on French data and the French situation, which should be somehow be mirrored in the title.

The abstract lacks a clear structure, as especially the method is missing. In a review, a specific study (UNICANCER-EVOLPEC study) should not be mentioned in the abstract. The introduction section needs references.

Line 120 ff: the URL of the Pipame report and others should be cited as a reference, not in the text.

Figure 1 is not very comprehensible. It is not whether all listed clear e-health tools can be used for all applications in oncology. The respective explanation given in line 123 is not very helpful to understand the figure. Also, the figure seems to be of low resolution, which also applies to figure 2.

Author Response

Round  2

Reviewer 1 Report

The new submitted version contains satisfactory changes, responding to almost all the issues I posed in my previous review. I appreciated in particular the effort of an update in reviewing the more recent literature.

I just have still one important request and some very minor comments.

First the request: although the work is well done, since it is a review, it may be even more clear and straightforward if the authors introduce a synoptic table, reporting the various studies in a compact manner (i.e. type of study, type of disease, number of subjects, type of e-health systems/devices used, brief results, possible comments).

Then there are four minor comments:

Line 29: “but also for health care..” -> “but also for caregivers (formal and informal) and health care..”

Line 45: ”staying at home” ->  ”staying at home (remotely supervised)”

Line 376: “in accessing the Internet” -> “in accessing the Internet or do not have the needed broadband connections (i.e. for multimedia communication or large file transfer)”

Line 489: “not only for patients” -> “not only for patients, but also for caregivers (formal and informal) and …”

Author Response

The new submitted version contains satisfactory changes, responding to almost all the issues I posed in my previous review. I appreciated in particular the effort of an update in reviewing the more recent literature.

We thank the reviewer for his/her positive comments.

I just have still one important request and some very minor comments.

1/ First the request: although the work is well done, since it is a review, it may be even more clear and straightforward if the authors introduce a synoptic table, reporting the various studies in a compact manner (i.e. type of study, type of disease, number of subjects, type of e-health systems/devices used, brief results, possible comments).

Even if the paper is now a Letter, rather than a Review (as suggested by the Academic Editor and Reviewer 2), we agree that such a table may be interesting for the reader. We have thus included a table (Table 1) summarizing some main studies published to date or ongoing in the field.

2/ Then there are four minor comments:

Line 29: “but also for health care..” -> “but also for caregivers (formal and informal) and health care..”

Ok. We have replacedbut also for health care,…” by “but also for caregivers (formal and informal) and health care,…”

Line 45: ”staying at home” ->  ”staying at home (remotely supervised)”

Ok. We have replacedstaying at home” by “staying at home (remotely supervised).”

Line 376: “in accessing the Internet” -> “in accessing the Internet or do not have the needed broadband connections (i.e. for multimedia communication or large file transfer)”

OK. We have replacedin accessing the Internet” by “in accessing the Internet or do not have the needed broadband connections (i.e. for multimedia communication or large file transfer)”.

Line 489: “not only for patients” -> “not only for patients, but also for caregivers (formal and informal) and …”

OK. We have replacednot only for patients” by “not only for patients, but also for caregivers (formal and informal) and …

Reviewer 2 Report

Although the authors provided a revised version which addressed some of the issues raised, I still have objectives regarding the method, as the authors rather present a letter or comment than a (systematic?) review, as guidelines for how to conduct a review are not followed and the method is not reported.

The figures are still of low resolution. Also, I would recommend explaining the abbreviation in the figure legend of figure 2.

Although the authors provided a revised version which addressed some of the issues raised, I still have objectives regarding the method, as the authors rather present a letter or comment than a (systematic?) review, as guidelines for how to conduct a review are not followed and the method is not reported.

Author Response

As suggested by this Reviewer and your Academic Editor, the paper is now presented as a Letter. The term “Review” has been deleted from the Abstract, and the term “comment” has been included. We have also added the following sentence at the end of Introduction: “In this Letter, we depict the current landscape of e-medicine in the field of cancer, and comment on the potential benefits and limits related to the use of e-health in the care management of cancer patients.”

The figures are still of low resolution. Also, I would recommend explaining the abbreviation in the figure legend of figure 2.

We apologize for this issue of low resolution. In fact, when we had previously submitted the Revised version, we had deleted with the Word changes-tracking system the low-resolution figures from the manuscript as suggested by Reviewer 2. However, they remained apparent because we had submitted the version with tracked changes and had not incorporated in the text the high-resolution figures that we had submitted as separate pdf files; Now, we have inserted the adequate figures directly in the text.

The SWOT abbreviation has been explained in the legend of Figure 2.